# Identification of a GrgA-Euo-HrcA Transcriptional Regulatory Network in *Chlamydia*

Wurihan Wurihan,[a] Yi Zou,[b] Alec M. Weber,[a] Korri Weldon,[b] Yehong Huang,[a,d] Xiaofeng Bao,[e] Chengsheng Zhu,[f] Xiang Wu,[d] Yaqun Wang,[g] Zhao Lai,[b,c] Huizhou Fan[a]

aDepartment of Pharmacology, Robert Wood Johnson Medical School, Rutgers University, Piscataway, New Jersey, USA
bGreehey Children's Cancer Research Institute, University of Texas Health San Antonio, San Antonio, Texas, USA
cDepartment of Molecular Medicine, University of Texas Health San Antonio, San Antonio, Texas, USA
dDepartment of Parasitology, Central South University Xiangya Medical School, Changsha, Hunan, China
eDepartment of Pharmacology, School of Pharmacy, Nantong University, Nantong, China
fDepartment of Microbiology and Biochemistry, School of Environmental and Biological Sciences, Rutgers University, New Brunswick, New Jersey, USA
gDepartment of Biostatistics, School of Public Health, Rutgers University, New Brunswick, New Jersey, USA

**ABSTRACT** *Chlamydia trachomatis* is an obligate intracellular bacterium whose unique developmental cycle consists of an infectious elementary body and a replicative reticulate body. Progression of this developmental cycle requires temporal control of the transcriptome. In addition to the three chlamydial sigma factors ($\sigma^{66}$, $\sigma^{28}$, and $\sigma^{54}$) that recognize promoter sequences of genes, chlamydial transcription factors are expected to play crucial roles in transcriptional regulation. Here, we investigate the function of GrgA, a *Chlamydia*-specific transcription factor, in *C. trachomatis* transcriptomic expression. We show that 10 to 30 min of GrgA overexpression induces 13 genes, which likely comprise the direct regulon of GrgA. Significantly, $\sigma^{66}$-dependent genes that code for two important transcription repressors are components of the direct regulon. One of these repressors is Euo, which prevents the expression of late genes during early phases. The other is HrcA, which regulates molecular chaperone expression and controls stress response. The direct regulon also includes a $\sigma^{28}$-dependent gene that codes for the putative virulence factor PmpI. Furthermore, overexpression of GrgA leads to decreased expression of almost all tRNAs. Transcriptomic studies suggest that GrgA, Euo, and HrcA have distinct but overlapping indirect regulons. These findings, together with temporal expression patterns of *grgA*, *euo*, and *hrcA*, indicate that a transcriptional regulatory network of these three transcription factors plays critical roles in *C. trachomatis* growth and development.

**IMPORTANCE** *Chlamydia trachomatis* is the most prevalent sexually transmitted bacterial pathogen worldwide and is a leading cause of preventable blindness in underdeveloped areas as well as some developed countries. *Chlamydia* carries genes that encode a limited number of known transcription factors. While Euo is thought to be critical for early chlamydial development, the functions of GrgA and HrcA in the developmental cycle are unclear. Activation of *euo* and *hrcA* immediately following GrgA overexpression indicates that GrgA functions as a master transcriptional regulator. In addition, by broadly inhibiting tRNA expression, GrgA serves as a key regulator of chlamydial protein synthesis. Furthermore, by upregulating *pmpI*, GrgA may act as an upstream virulence determinant. Finally, genes coregulated by GrgA, Euo, and HrcA likely play critical roles in chlamydial growth and developmental control.

**KEYWORDS** CT504, CTL0766, *Chlamydia*, Euo, GrgA, HrcA, transcription factors, transcriptional regulatory network

Address correspondence to Huizhou Fan, huizhou.fan@rutgers.edu.

This work shows that GrgA functions as a master transcription factor in Chlamydia. GrgA activates two other TF-encoding genes, Euo and HrcA. The GrgA-Euo-HrcA network likely plays critical roles in chlamydial growth and development.

Chlamydiae are obligate intracellular Gram-negative bacteria with a unique developmental cycle characterized by two cellular forms (1). The small form, known as the elementary body (EB), is capable of short-term extracellular survival but incapable of proliferation. After entering the host cell through endocytosis, the EB differentiates into a larger form known as the reticulate body (RB) inside a vacuole termed the inclusion. The RB replicates with a doubling time of about 2 h (2–4). Around 24 h, some RBs start to redifferentiate back into EBs while others continue to proliferate (3, 5). At the end of the cycle, both infectious EBs and noninfectious RBs are released from host cells through either cell lysis or extrusion of entire inclusions (6, 7).

Expression of the chlamydial transcriptome is developmentally regulated. Previous cDNA microarray studies (8, 9) enumerate four successive stages of the *Chlamydia trachomatis* developmental cycle. The immediate early stage is defined as the first hour when EBs are inside nascent inclusions near the plasma membrane. Only a small number of presumably crucial genes are transcribed in this stage to establish an intracellular niche that enables EB survival, RB development, and eventual delivery of the inclusion to a perinuclear region. During the subsequent early stage, an additional number of genes are transcribed to complete the conversion of EBs into RBs. Midcycle commences upon the completion of EB-to-RB conversion and ends when RBs start to differentiate back into EBs. Almost all genes are transcribed during this stage. Last, transcription of a smaller set of genes is initiated and/or upregulated before and during the late stage.

*C. trachomatis* carries genes that encode three sigma factors (10, 11), which guide the RNA polymerase to different promoters (12, 13). The vast majority of *C. trachomatis* promoters are $\sigma^{66}$ dependent, while some late genes may possess $\sigma^{28}$ or $\sigma^{54}$ promoters (14–19). A few late genes have tandem promoters for $\sigma^{66}$ and an alternative $\sigma$ (14). Consistent with their roles in the developmental cycle, expression of the three sigma factors is also temporally regulated (8, 9, 20). $\sigma^{66}$ RNA is detected by microarray as early as 3 h postinoculation (hpi), whereas $\sigma^{28}$ and $\sigma^{54}$ mRNAs are not detected until 8 hpi (8).

*C. trachomatis* carries genes that encode nearly 20 known transcription factors (TFs), which regulate transcription from different promoters (10, 11, 21–34). To date, only two TFs are known to regulate the chlamydial developmental cycle: Euo and CtcC. Euo is produced immediately after EBs enter host cells and occupies $\sigma^{66}$- and/or $\sigma^{28}$-dependent late gene promoters to suppress their transcription (8, 18, 35–37). CtcC functions as an activator of the $\sigma^{54}$-RNA polymerase (RNAP) holoenzyme, which regulates RB-to-EB differentiation (14, 38). CtcC also targets a gene that encodes the transcription repressor HrcA, whose role in chlamydial development remains unclear. While two microarray studies showed that *hrcA* is a late gene, one also showed high RNA levels of *grpE* and *dnaK*, which are in the same operon as *hrcA*, at 3 hpi (8).

GrgA is the most recently discovered chlamydial TF (39). Identified via promoter DNA pulldown, GrgA physically interacts with $\sigma^{66}$ and $\sigma^{28}$ and activates transcription from both $\sigma^{66}$- and $\sigma^{28}$-dependent promoters *in vitro* (39–41). Transcriptomic studies presented in this report define the regulon of GrgA and identify a GrgA-directed transcriptional regulatory network (TRN). Similar to *euo*, expression analysis reveals that *hrcA* is also an immediate early gene and suggests the possibility that the GrgA protein prepackaged within EBs plays a critical role in the activation of *euo*, *hrcA*, and other genes after the EB enters the host cell.

## RESULTS

**GrgA overexpression-mediated global transcriptomic changes include upregulated *euo*, *hrcA*, and *pmpl* expression and downregulated tRNA expression.** We performed transcriptome sequencing (RNA-seq) analysis for *C. trachomatis* L2 (CtL2) transformed with an anhydrotetracycline (ATC)-inducible GrgA expression plasmid (4) to determine the role of GrgA in *C. trachomatis* transcriptomic expression. In our pilot RNA-seq experiments, GrgA transformants (i.e., CtL2/GrgA) grown in mouse fibroblast L929 cells were treated with or without ATC within two time periods: 12 to 16 hpi and 17 to 21 hpi (*n* = 1 per experimental condition). The rates of reads mapped

to the CtL2 genome of samples prepared at 16 hpi for noninduced and ATC-induced cultures were 13.0% and 16.8%, respectively (see Data Set S1A in the supplemental material). The numbers of mapped reads resulted in 135-fold and 164-fold CtL2 genome coverage, respectively. As expected, the rates of reads mapped to the CtL2 genome of samples prepared at 21 hpi were 3- to 4-fold higher (i.e., 44.1% and 52.6%) (Data Set S1A). Conversely, the rates of reads mapped to the mouse genome of samples prepared at 16 hpi were higher than the rates of those at 21 hpi (data not shown). The total mapped rate (i.e., the chlamydial and murine mapping rates combined) ranged from 95.1% to 96.6% (data not shown).

With the exception of 5S and 16S rRNAs, which were depleted prior to library construction, we detected RNAs of all but seven chlamydial genes at 16 hpi despite the notably lower reads at this time point. In both sets of experiments, RNAs of two transcription repressor-encoding genes *euo* and *hrcA* were noticeably increased in ATC-induced cultures. For the 12 to 16 hpi induction, *euo* and *hrcA* increased by 3.1- and 2.8-fold, respectively. For the 17 to 21 hpi induction, they increased by 3.4- and 1.9-fold, respectively (Data Set S1B).

Subsequent RNA-seq studies were conducted with samples harvested from infected L929 cells at 16 hpi. We chose this time point first because it corresponds to the mid-log phase of RB replication, and we are most interested in uncovering how GrgA mediates RB growth regulation, and additionally because the genome coverages in the above pilot study with samples prepared at 16 hpi exceeded the recommended coverage for bacteria with small genomes (42). We repeated RNA-seq analysis with ATC-treated biological replicates for the 12 to 16 hpi time period to generate data that could be analyzed statistically. Consistent with our previous two RNA-seq studies with a single sample per condition (Data Set S1), transcripts of both *euo* and *hrcA* also increased significantly in response to ATC treatment in the duplicated experiments. *euo* increased by 3.3-fold, the highest increase not including ATC-induced increase in the *grgA* RNA (Data Set S2B). *hrcA* increased by 2.1-fold, the fifth largest increase. Transcripts of 81 other genes also increased by ≥1.33-fold with an adjusted $P$ value of <0.05 (Data Set S2B). We chose this relatively low cutoff to increase the chance of identifying biological targets of GrgA, a strategy used by previous RNA-seq studies (43–45). Among the 81 other upregulated genes is *pmpI*, which encodes a polymorphic protein in the outer membrane complex that is a putative virulence factor (46, 47). *pmpI* was upregulated by 2.5-fold, second only to *euo* (Data Set S2B). Retrospective analysis showed that *pmpI* was also upregulated by 2.7-fold and 2.5-fold in the aforementioned pilot study with ATC treatment from 12 to 16 hpi and from 17 to 21 hpi, respectively, without biological replicates (Data Set S1B). Collectively, *euo*, *hrcA*, *pmpI*, and the remaining 80 genes upregulated by GrgA fell into 14 different functional groups, as will be described in TRN analysis.

Transcripts of 78 genes decreased by 1.33-fold with an adjusted $P < 0.05$ in the biologically duplicated RNA-seq study (Data Set S2B). Of these 78 genes, 28 were tRNA genes. Only 9 of the 37 tRNAs were not significantly downregulated (Table 1). Retrospective analysis showed that 26 tRNAs were also decreased by more than 1.33-fold in the pilot experiment with ATC treatment from 12 to 16 hpi without biological replicates (Data Set S1B).

Taken together, these RNA-seq studies performed for cultures with 4-h ATC treatment consistently demonstrated upregulated *euo*, *hrcA*, and *pmpI* expression in response to GrgA overexpression. These findings suggest the possibility that GrgA functions as an activator of *euo*, *hrcA*, and *pmpI*. In addition, the studies indicate that GrgA regulates expression of numerous other genes either directly or indirectly.

**Upregulation of *euo* and *hrcA* but not *pmpI* depends on $\sigma^{66}$ binding of GrgA.** Previously published *in vitro* studies showed that GrgA activates both $\sigma^{66}$-dependent transcription and $\sigma^{28}$-dependent transcription by physically interacting with $\sigma^{66}$ and $\sigma^{28}$, respectively (39, 40). To determine the roles of GrgA's interactions with $\sigma^{66}$ and $\sigma^{28}$ in GrgA overexpression-induced transcriptomic changes, we performed RNA-seq

mSystems®

**TABLE 1** tRNAs downregulated by 4-h GrgA overexpression[a]

| Category and locus tag | tRNA | Fold change | Adjusted P value |
|---|---|---|---|
| ≥1.33-fold decrease (adjusted $P < 0.05$) | | | |
| CTL_t02 | tRNA-Ala2 | −2.0448124 | 0.00027 |
| CTL_t03 | tRNA-Arg | −1.6889674 | 0.00961 |
| CTL_t06 | tRNA-Asn | −1.9534442 | 9.31E−08 |
| CTL_t07 | tRNA-Asp | −1.7031689 | 3.33E−05 |
| CTL_t08 | tRNA-Cys | −2.0778794 | 1.74E−05 |
| CTL_t09 | tRNA-Gln | −2.1846452 | 1.04E−09 |
| CTL_t10 | tRNA-Glu | −1.6229639 | 6.69E−05 |
| CTL_t11 | tRNA-Gly | −1.8190017 | 5.65E−05 |
| CTL_t12 | tRNA-Gly2 | −1.8146205 | 5.52E−05 |
| CTL_t14 | tRNA-Ile | −1.6209525 | 0.000925 |
| CTL_t15 | tRNA-Leu4 | −1.4443658 | 0.023179 |
| CTL_t16 | tRNA-Leu3 | −1.7377065 | 1.01E−05 |
| CTL_t17 | tRNA-Leu | −2.1501725 | 4.46E−11 |
| CTL_t18 | tRNA-Leu5 | −1.9135418 | 1.74E−05 |
| CTL_t19 | tRNA-Leu2 | −1.9158336 | 4.84E−08 |
| CTL_t20 | tRNA-Lys | −1.6740043 | 0.00028 |
| CTL_t22 | tRNA-Met | −1.5424232 | 0.00184 |
| CTL_t24 | tRNA-Phe | −2.1952479 | 8.32E−05 |
| CTL_t27 | tRNA-Ser3 | −1.7954104 | 1.10E−05 |
| CTL_t28 | tRNA-Ser4 | −1.7413659 | 5.52E−05 |
| CTL_t29 | tRNA-Ser | −1.8662364 | 4.16E−07 |
| CTL_t30 | tRNA-Ser2 | −2.1874562 | 6.90E−10 |
| CTL_t31 | tRNA-Thr2 | −1.6366313 | 5.66E−05 |
| CTL_t32 | tRNA-Thr | −2.1215215 | 5.26E−08 |
| CTL_t33 | tRNA-Thr3 | −1.4370903 | 0.00705 |
| CTL_t34 | tRNA-Trp | −1.3942998 | 0.03454 |
| CTL_t35 | tRNA-Tyr | −1.6703612 | 1.47E−05 |
| CTL_t37 | tRNA-Val | −1.8533329 | 0.000204 |
| | | | |
| <1.33-fold change or adjusted $P > 0.05$ | | | |
| CTL_t01 | tRNA-Ala | −0.878068 | 0.691436 |
| CTL_t04 | tRNA-Arg3 | −1.9399986 | 0.102252 |
| CTL_t05 | tRNA-Arg2 | −1.7983519 | 0.132967 |
| CTL_t13 | tRNA-His | −2.1978167 | 0.074222 |
| CTL_t21 | tRNA-Met2 | −1.1130862 | 0.762719 |
| CTL_t23 | tRNA-Met3 | −1.6251197 | 0.066761 |
| CTL_t25 | tRNA-Pro | −1.5097685 | 0.051475 |
| CTL_t26 | tRNA-Pro2 | −1.8040708 | 0.175928 |
| CTL_t36 | tRNA-Val2 | −0.9681379 | 1 |

[a]Values were extracted from Data Set S2 obtained with biological duplicates.

for CtL2/GrgAΔσ⁶⁶BD and CtL2/GrgAΔσ²⁸BD, which are CtL2 transformed with inducible expression plasmids for GrgA that lacked either the $\sigma^{66}$-binding domain (GrgAΔσ⁶⁶BD) or the $\sigma^{28}$-binding domain (GrgAΔσ²⁸BD), respectively. In ATC-treated CtL2/GrgAΔσ⁶⁶BD cultured in L929 cells, upregulation of only a single gene was statistically significant, while downregulation of only four genes was statistically significant (Fig. 1 and Data Set S2C), even though all forms of transformed GrgA are expressed upon ATC induction (Fig. S1). In ATC-treated CtL2/GrgAΔσ²⁸BD, the numbers of upregulated and downregulated genes were both higher than those of ATC-treated CtL2/GrgA (Fig. 1 and Data Set S2D).

The sole gene upregulated by GrgAΔσ⁶⁶BD overexpression was *pmpI*, which was also upregulated by overexpression of both full-length GrgA and GrgAΔσ²⁸BD (Fig. 1A and Data Set S2E). Noticeably, *pmpI* expression increased after full-length GrgA overexpression and GrgAΔσ⁶⁶BD overexpression by a similar magnitude (2.5-fold versus 1.7-fold). These increases were substantially higher than the 40% increase observed after GrgAΔσ²⁸BD overexpression (Data Set S2E). Fifty-one genes, including *euo* and *hrcA*, were upregulated following both full-length GrgA overexpression and GrgAΔσ²⁸BD overexpression (Fig. 1A and Data Set S2E).

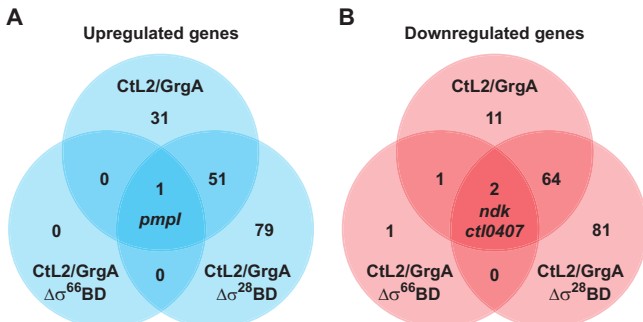

**FIG 1** Venn diagrams showing numbers of up- and down-regulated genes detected in ATC-treated *C. trachomatis* L2 (CtL2) transformants of full-length GrgA (CtL2/GrgA), GrgAΔσ⁶⁶BD (CtL2/GrgAΔσ⁶⁶BD), and GrgAΔσ²⁸BD (CtL2/GrgAΔσ²⁸BD). CtL2/GrgA-, CtL2/GrgAΔσ⁶⁶BD-, or CtL2/GrgAΔσ²⁸BD-infected L929 cells in biological duplicates were treated with 10 nM ATC at 12 hpi or left untreated. Cultures were terminated at 16 hpi and processed for RNA-seq analysis as described in Materials and Methods. Venn diagrams were derived from RNA-seq data presented in Data Set S2 in the supplemental material. Identities of genes commonly upregulated or downregulated by all three GrgA constructs are shown. Genes commonly upregulated or downregulated by any two of the constructs are listed in Data Set S2E. Up- or down-regulation was defined as a ≥ 1.33-fold change with an adjusted $P < 0.05$.

Two of the four genes downregulated by GrgAΔσ⁶⁶BD overexpression were *ndk* (nucleotide diphosphate kinase) and *ctl0407* (lipoprotein releasing system ABC transporter ATP-binding protein), which were also downregulated by overexpression of both full-length GrgA and GrgAΔσ²⁸BD (Fig. 1B and Data Set S3E). *dppD* (oligopeptide ABC transporter ATP-binding protein) is the third GrgAΔσ⁶⁶BD-downregulated gene, which was also downregulated by full-length GrgA. In total, 66 genes were downregulated by both full-length GrgA overexpression and GrgAΔσ²⁸BD overexpression (Fig. 1B). Remarkably, 28 of these 66 genes encode tRNAs (Data Set S2E).

Taken together, comparative transcriptomic analysis (Fig. 1 and Data Set S2) suggests that nearly all transcriptomic changes (including upregulation of *euo* and *hrcA*) induced by GrgA overexpression depend on GrgA binding of σ⁶⁶. In contrast, fewer changes (e.g., upregulation of *pmpI*) depend on GrgA binding of σ²⁸. However, overexpression of GrgAΔσ²⁸BD can induce additional transcriptomic changes not seen with full-length GrgA overexpression. Therefore, the σ²⁸-binding domain may restrict upregulation of certain σ⁶⁶-dependent genes *in vivo*.

***euo*, *hrcA*, and 11 other genes are upregulated immediately following GrgA overexpression.** All three RNA-seq studies presented above were performed in cultures with and without 4-h ATC treatment. To identify genes directly targeted by GrgA, we determined transcriptomic kinetics by extracting RNA at 16 hpi from GrgA transformant-infected L929 cultures with 10 nM ATC treatment initiated at 15.5, 15, or 14 hpi or without ATC treatment. This treatment schedule allowed for 0.5, 1, or 2 h of GrgA overexpression, respectively. Using Gaussian mixture models (48), the entire transcriptome was divided into six groups based on changes in the expression kinetics of individual genes (Fig. 2 and Data Set S3E). Group A contains six genes whose functions are presented in Fig. 3A. These six genes were upregulated as early as 0.5 h after induction (Fig. 2A), although the upregulation at 0.5 h was statistically significant for only four of the six genes. The four statistically significantly upregulated genes were *euo*, *pmpI*, *aroC*, and *lplA*. These genes could be direct targets of GrgA. A total of 174 genes, including *hrcA*, were upregulated by 1 h (Fig. 2B). Most genes of this large group may be secondary or indirect targets of GrgA. Expression of 444 genes remained relatively constant (Fig. 2C) and are therefore unlikely targets of GrgA considering GrgA acts as a transcription activator (39, 40). Expression of the remaining genes decreased by various degrees (Fig. 2D to F), likely in response to expression changes of the primary and/or secondary targets.

A series of quantitative reverse transcription-PCR (qRT-PCR) experiments were performed to validate the RNA-seq data for all genes upregulated by 0.5-h ATC induction and selected genes upregulated by 1-h ATC induction because they might be directly

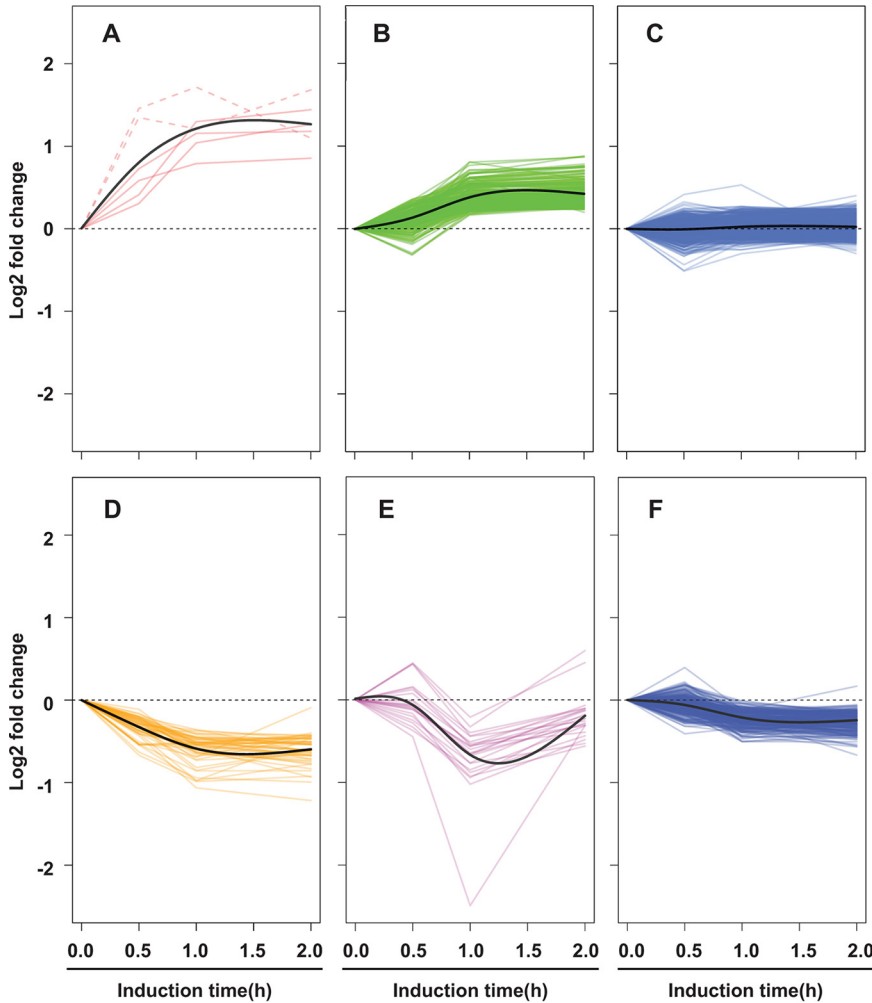

**FIG 2** Temporal patterns of transcriptomic changes induced by GrgA overexpression. CtL2/GrgA-infected L929 cultures in biological triplicates were treated with 10 nM ATC at 14, 15, or 15.5 hpi or left untreated. Cultures were terminated at 16 hpi and processed for RNA-seq analysis. *C. trachomatis* genes were clustered into six groups of temporal expression change patterns by analyzing RNA-seq data presented in Data Set S3. (A) Six target genes were increased by 0.5 h of ATC induction. RNAs whose increases were statistically significant (i.e., with an adjusted $P < 0.05$) are shown in solid lines. RNAs that increased with an adjusted $P > 0.05$ by 0.5 h are shown in dashed lines. (B) RNAs of 175 genes are stimulated by GrgA overexpression only after 1 h of ATC induction. (C) RNAs of 444 genes remained relatively constant. (D to F) Genes are downregulated following different kinetics. In panels A to F, solid black lines are trend lines in the respective groups.

targeted by GrgA. Functions of genes further analyzed by qRT-PCR are shown in Fig. 3A. Among the six genes upregulated by 0.5 h in RNA-seq (Fig. 2A and Data Set S3), *euo*, *pmpI*, *ctl0758*, and *ctl0418* are nonoperon genes (Fig. 3B). Results showed that expression of *euo* and *pmpI* increased about 2- and 3-fold, respectively, at 10 min after induction, and by more than 3- and 4-fold, respectively, at 30 min (Fig. 3B). Smaller but significant increases were detected for the mRNA of *ctl0758* from 10 to 30 min (Fig. 3B). However, a 57% increase in *ctl0418* expression was not detected until 30 min, and this increase was not statistically significant (Fig. 3B).

Based on the genome organization (10, 11) and results of a study on *C. trachomatis* transcriptional start sites (49), *lplA* and *ctl0536* lie within one operon (Fig. 3C), while *aroC* and three other genes lie within another (Fig. 3D). Noticeably, RNA-seq did not detect concordant upregulation of genes cotranscribed with *lplA* and *aroC* (Data Set S3). Therefore, we also included a cotranscribed gene from the same operon in our qRT-PCR analysis. This analysis shows that the expression trends of both *lplA* and

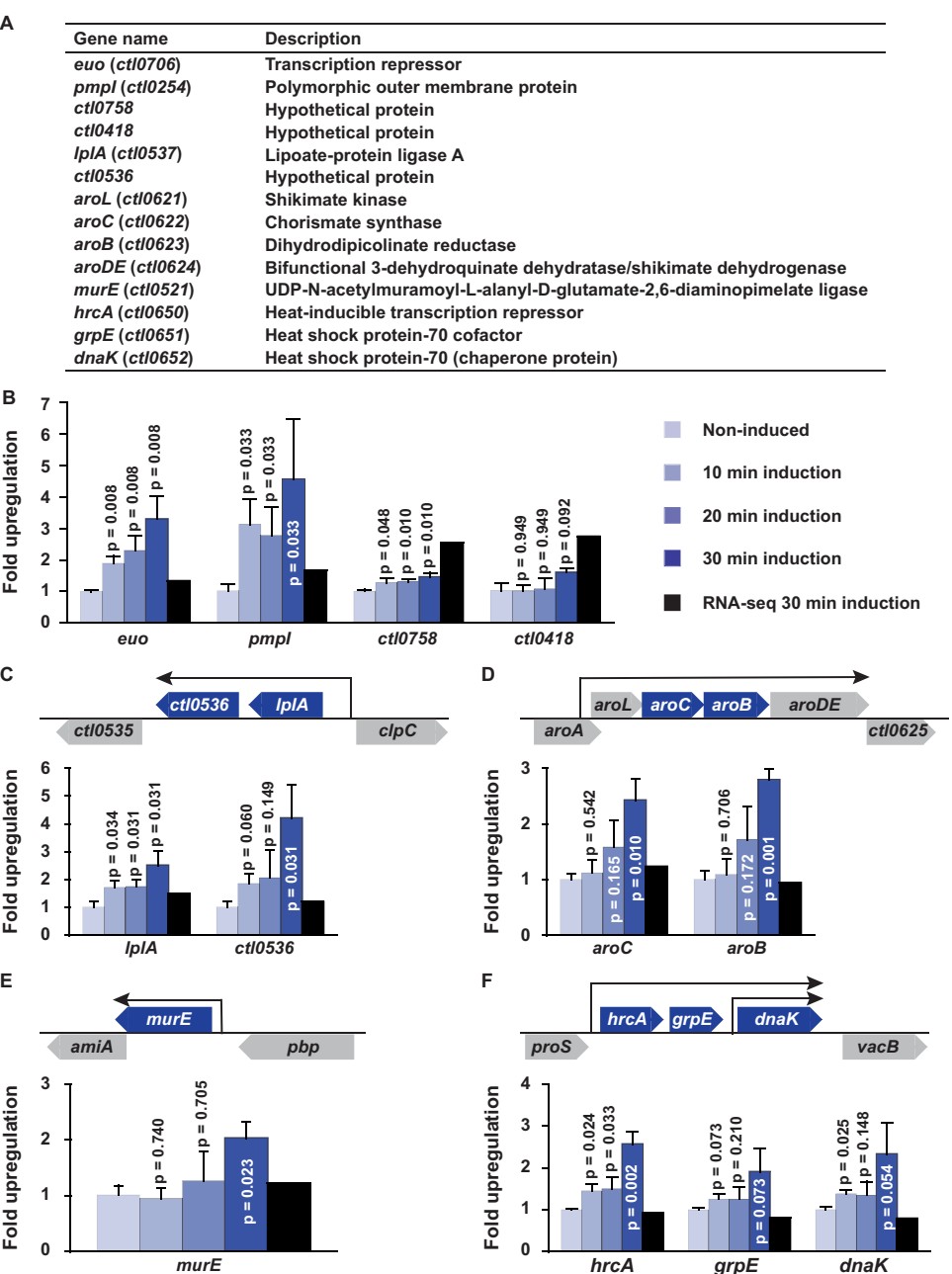

**FIG 3** qRT-PCR detection or confirmation of genes upregulated by GrgA within 10 to 30 min of ATC treatment. CtL2/GrgA-infected L929 cultures in biological triplicates were treated with 10 nM ATC for 10, 20, or 30 min or left untreated. Cultures were terminated at 16 hpi. The levels of expression of individual genes were determined using qRT-PCR. (A) Descriptions of the genes studied in this figure. (B) Three of four nonoperon genes from Fig. 2A showed increased mRNA levels by 10 to 30 min. The color code to the right of the graph applies to panels B to F. (C) Expression of *lplA* and its cotranscribed gene *ctl0536* was increased at 10 and 30 min, respectively. (D) Transcripts of both *aroC* and its cotranscribed gene *aroB* were increased by 30 min. *aroL* and *aroDE* from the same operon were not analyzed. (E) Expression of the nonoperon gene *murE* was significantly increased at 30 min. (F) Transcripts of *hrcA* and its cotranscribed genes *grpE* and *dnaK* were increased by 10 min, although the increases in *grpE* were less pronounced with only trending significant adjusted *P* values. In panels B to F, data are averages plus standard deviations (error bars). All *P* values were adjusted for multiple comparison. Upregulation detected by RNA-seq at 0.5 h after ATC treatment (Data Set S3) is presented as a black bar for comparison.

*ctl0536* (Fig. 3C) were similar to those of *euo* and *pmpl* (Fig. 3B) with a significant increase for *lplA* and a trending significant increase for *ctl0536* starting at 10 min. Statistically significant increases were also found for *aroC* and *aroB* by 30 min (Fig. 3D). These results validate *lplA* and *aroC* as genes immediately induced by GrgA and further

suggest that *aroL* and *aroDE*, which lie within the same operon as *aroB* and *aroC*, are also induced by GrgA though we did not perform qRT-PCR for either *aroL* or *aroDE*.

The apparent higher detection sensitivity of qRT-PCR analysis over RNA-seq (Fig. 3C and D) prompted its use to determine whether certain genes whose expression did not increase until 1 h might actually be upregulated at 30 min or even earlier. Our inclusion criteria to select genes for qRT-PCR analysis were (i) a read increase with adjusted $P < 0.05$, and (ii) at least one FPKM (fragment per kilobase of gene length per million reads of the library) value $> 900$ for induced samples. Twelve genes met these criteria (Data Set S3). An exception was made for *hrcA* (FPKM $= 369 \pm 66$ [mean $\pm$ standard deviation {SD}]), because HrcA is a TF whose RNA was consistently increased by GrgA overexpression in all previous RNA-seq studies conducted with 4-h ATC induction (Fig. 1 and Data Sets S1 and 2). qRT-PCR analysis detected apparently increased levels for all mRNAs analyzed; however, only the increases in *murE* and *hrcA* were statistically significant (adjusted $P < 0.05$). Further analysis confirmed that the *murE* RNA increased at 30 min but not 10 or 20 min after ATC induction (Fig. 3E), while *hrcA* RNA readily increased at even 10 min (Fig. 3F). *hrcA* is in the same operon as *grpE* (which encodes heat shock protein-70 cofactor) and *dnaK* (a protein chaperone gene), although *dnaK* also has an additional promoter for itself (50). Similar to the *hrcA* gene, *grpE* and *dnaK* showed increased expression starting at 10 min, although the increase in *grpE* was borderline significant ($P = 0.073$) (Fig. 3F).

In addition to analyzing expression at different times after 10 nM ATC induction, we investigated *euo* and *hrcA* expression in response to multiple low concentrations of ATC using qRT-PCR. This analysis demonstrated dose-dependent increases in both *euo* and *hrcA* expression as early as 30 min after ATC induction (Fig. S2). Taken together, the time- and dose-kinetic results presented in Fig. 3 and Fig. S2 demonstrate that four nonoperon genes (*euo*, *pmpI*, *murE*, and *ctl0758*) and nine additional genes (*ctl0536*, *lplA*, *aroL*, *aroC*, *aroB*, *aroDE*, *hrcA*, *grpE*, and *dnaK*) in three operons are upregulated by 10- to 30-min induction of GrgA overexpression. Most likely, these 13 genes comprise GrgA's direct regulon.

**GrgA stimulates transcription from *euo*, *hrcA*, and *pmpI* promoters *in vitro*.** The earlier obtained RNA-seq data from CtL2/GrgA, CtL2/GrgAΔ$\sigma^{66}$BD, and CtL2/GrgAΔ$\sigma^{28}$BD (Fig. 1 and Data Set S2) support the notion that GrgA activates genes with $\sigma^{66}$ promoters as well as genes with $\sigma^{28}$ promoters. To provide further evidence for this proposition, we searched for $\sigma^{66}$ and $\sigma^{28}$ promoter elements in the promoter regions of the four nonoperon genes and three operons that were upregulated by 30 min of GrgA overexpression (Fig. 3). Among the four nonoperon genes (i.e., *euo*, *pmpI*, *murE*, and *ctl0758*) and three operons (i.e., the *ctl0536-lplA* operon, *aroLCBDE* operon, and *hrcA-grpE-dnaK* operon) upregulated by GrgA overexpression at 10 to 30 min (Fig. 3), hexameric nucleotide sequences resembling the −10 elements of *C. trachomatis* $\sigma^{66}$ (51, 52) were readily recognized 6 to 9 bp upstream of the perspective transcription start sites previously identified for *euo*, *hrcA*, *murE*, and *aroL* (49, 51, 52), whereas hexameric nucleotide sequences resembling putative −35 elements of *C. trachomatis* $\sigma^{66}$ promoters (53) were readily found 17 or 18 nucleotides upstream of the putative −10 elements for *euo*, *hrcA*, and *aroL*. The core promoter sequences, including the putative −10 and −35 elements, the spacing sequences between the elements, and the transcription start sites of *euo* and *hrcA* are shown in Fig. 4A. In addition, an octameric nucleotide sequence resembling the −10 element of *C. trachomatis* $\sigma^{28}$ promoter (16, 51, 52) was readily found 8 nucleotides upstream of the transcription start site previously identified for *pmpI*, while another octameric nucleotide sequence resembling the −35 elements of *C. trachomatis* $\sigma^{28}$ promoters (51, 52, 54) was detected 11 nucleotides upstream of the −10 element (Fig. 4A). Next, we constructed three *in vitro* transcriptional reporter plasmids, of which two carried a putative $\sigma^{66}$ promoter of *euo* or *hrcA*. The third plasmid carried the putative $\sigma^{28}$ promoter of *pmpI*. All three plasmids were able to direct RNA synthesis, with increased transcripts detected in the presence of GrgA (Fig. 4B). These results confirm that *euo*, *hrcA*, and its cotranscribed genes *grpE*, *dnaK*, and *pmpI* are direct targets of GrgA. They provide further support for

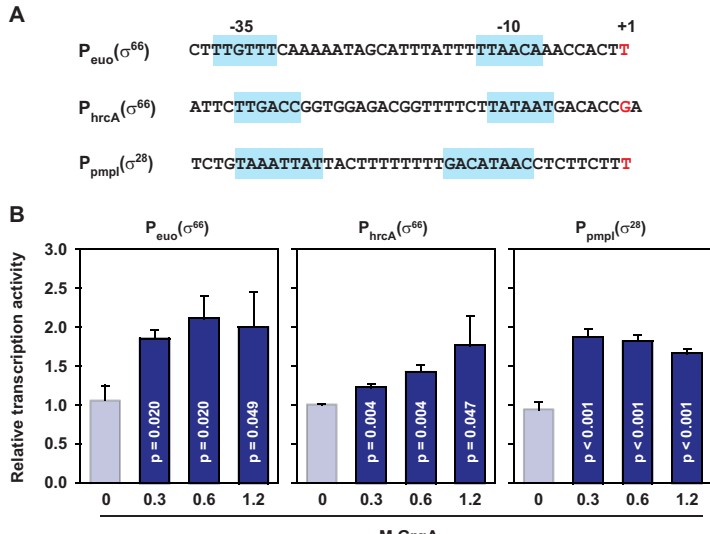

**FIG 4** Stimulation of transcription from *euo*, *hrcA*, and *pmpI* promoters by GrgA *in vitro*. (A) Core promoter sequences in the promoter fragments cloned into transcriptional reporter plasmids pMT1125-Peuo, pMT1125-PhrcA, and pMT1125-PpmpI($\sigma^{28}$) (Table 2). Nucleotides shown in red and marked with +1 signify the transcription start sites previously identified (49). −10 and −35 elements of $\sigma^{66}$ and $\sigma^{28}$ promoters are indicated by a light blue background. (B) Stimulation of transcription from indicated promoters by GrgA. Data are averages plus standard deviations of three independent experiments. *P* values were adjusted for multiple comparison.

the notion that GrgA activates transcription from not only the $\sigma^{66}$ promoters but also the $\sigma^{28}$ promoter in *C. trachomatis*.

**Identification of genes commonly regulated by GrgA, Euo, and HrcA.** The consistent upregulation of *euo* and *hrcA* by GrgA in both the transcriptomic kinetic studies (Fig. 3 and Data Set S3) and 4 h induction RNA-seq studies (Data Sets S1 and S2), as well as GrgA-mediated activation of transcription from the *euo* and *hrcA* promoters *in vitro* (Fig. 4), imply the existence of a GrgA-Euo-HrcA transcriptional regulatory network (TRN). As a first step to understand this TRN, we derived CtL2 transformants that carried an inducible plasmid to express either Euo (CtL2/Euo) or HrcA (CtL2/HrcA). RNA-seq studies were then conducted to identify transcriptomic changes in these transformants following ATC induction between 12 and 16 hpi. The processed RNA-seq data of these RNA-seq studies are presented in Data Set S4. qRT-PCR analysis was performed for five genes with significantly increased RNA-seq reads and five genes with significantly reduced RNA-seq reads from each transformant following ATC treatment. The qRT-PCR data (Data Set S4D) were consistent with the RNA-seq data (Data Set S4B and S4C).

We identified 27 genes that are commonly regulated in all three transformants (i.e., CtL2/GrgA, CtL2/Euo, and CtL2/HrcA) following ATC treatment (Fig. 5 and Data Set S4E). Eight of these genes were commonly upregulated (Fig. 5A and B), while 19 genes were commonly downregulated (Fig. 5C and D). Noticeably, four of the eight commonly upregulated genes encode proteins involved in DNA replication, while the remaining four encode proteins with various functions (Fig. 5B). Among the 19 commonly downregulated genes, 13 encode functionally known proteins, 5 encode hypothetical proteins, and 1 encodes a glutamic acid tRNA (Fig. 5D). Three (PPA, IspH, and FabI) of the 13 functionally known proteins are enzymes that catalyze metabolic reactions; another three (RpsA, RplM, and RpmB) are ribosomal proteins. OmpA and CTL0887 are the major and a minor component, respectively, of the outer membrane protein complex. At least three proteins are related to type III secretion (T3S): YajC is a T3S protein translocase, CTL0299 is a T3S chaperone, and CTL0874 is virulence factor secreted by the T3S system. Among the remaining two proteins, CTL0476 is an

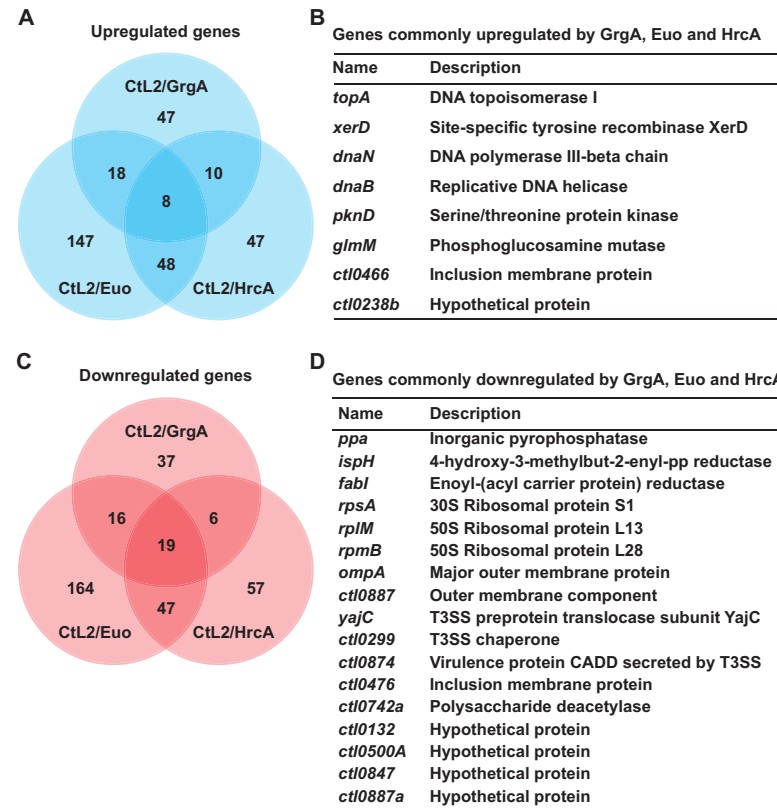

**A** Upregulated genes

CtL2/GrgA
47
18 | 10
8
147 | 48 | 47
CtL2/Euo | CtL2/HrcA

**B** Genes commonly upregulated by GrgA, Euo and HrcA

| Name | Description |
|---|---|
| *topA* | DNA topoisomerase I |
| *xerD* | Site-specific tyrosine recombinase XerD |
| *dnaN* | DNA polymerase III-beta chain |
| *dnaB* | Replicative DNA helicase |
| *pknD* | Serine/threonine protein kinase |
| *glmM* | Phosphoglucosamine mutase |
| *ctl0466* | Inclusion membrane protein |
| *ctl0238b* | Hypothetical protein |

**C** Downregulated genes

CtL2/GrgA
37
16 | 6
19
164 | 47 | 57
CtL2/Euo | CtL2/HrcA

**D** Genes commonly downregulated by GrgA, Euo and HrcA

| Name | Description |
|---|---|
| *ppa* | Inorganic pyrophosphatase |
| *ispH* | 4-hydroxy-3-methylbut-2-enyl-pp reductase |
| *fabI* | Enoyl-(acyl carrier protein) reductase |
| *rpsA* | 30S Ribosomal protein S1 |
| *rplM* | 50S Ribosomal protein L13 |
| *rpmB* | 50S Ribosomal protein L28 |
| *ompA* | Major outer membrane protein |
| *ctl0887* | Outer membrane component |
| *yajC* | T3SS preprotein translocase subunit YajC |
| *ctl0299* | T3SS chaperone |
| *ctl0874* | Virulence protein CADD secreted by T3SS |
| *ctl0476* | Inclusion membrane protein |
| *ctl0742a* | Polysaccharide deacetylase |
| *ctl0132* | Hypothetical protein |
| *ctl0500A* | Hypothetical protein |
| *ctl0847* | Hypothetical protein |
| *ctl0887a* | Hypothetical protein |
| *ctl0895* | Hypothetical protein |
| *tRNA-Glu* | tRNA for glutamic acid |

**FIG 5** GrgA-, Euo-, and HrcA-coregulated genes. RNA-seq data generated from biologically duplicated cultures of CtL2/GrgA (Data Set S2B), CtL2/Euo (Data Set S4B), and CtL2/HrcA (Data Set S4C) treated with 10 nM ATC from 12 to 16 hpi were analyzed to reveal genes coregulated by GrgA, Euo, and HrcA. (A) Venn diagrams showing numbers of genes upregulated by overexpression of each of the 3 TFs as shown in Data Set S4E. (B) List of genes commonly upregulated by GrgA, Euo, and HrcA overexpression. Note that genes commonly upregulated by two TFs are shown in Data Set S4E. (C) Venn diagrams showing numbers of genes downregulated by overexpression of each of the three TFs as shown in Data Set S4E. (D) List of genes commonly downregulated by GrgA, Euo, and HrcA overexpression. Note that genes commonly downregulated by two TFs are shown in Data Set S4E. Up- or downregulation was defined as a ≥1.33-fold change with an adjusted $P < 0.05$.

inclusion membrane protein secreted through a yet-undefined mechanism; CTL0742a is a polysaccharide deacetylase. These commonly upregulated and downregulated genes may serve as active regulators of RB growth.

**GrgA-, Euo-, and HrcA-controlled transcriptional regulatory network.** Next, we extracted differentially regulated genes from the RNA-seq data sets generated from experiments with biological duplicates or triplicates (Data Sets S2 to S4). The extracted RNA-seq data (Data Set S5) were combined with qRT-PCR data (Fig. 3 and Fig. S3) to elucidate GrgA TRNs. We used STRING to generate the TRNs because STRING can integrate the GrgA TRNs with previously identified functional association networks (55). A GrgA TRN within 10 to 30 min of ATC treatment was developed using qRT-PCR data obtained from CtL2/GrgA cultures treated with ATC for 10 to 30 min (Fig. 3). In this network, GrgA upregulated 13 genes and downregulated 8 genes (Fig. 6A). Among the 13 GrgA-upregulated genes, *euo*, *hrcA*, and *pmpI* were demonstrated as direct target genes of GrgA by *in vitro* transcription from their perspective promoters (Fig. 4). In addition, *grpE* and *dnaK*, both with functions in protein folding and stress response, are also considered direct targets of GrgA because they are cotranscribed with *hcrA* (49, 50, 56–59) and upregulated following GrgA overexpression (Fig. 3). As for the remaining eight upregulated genes, which are likely GrgA targets, four genes in the *aroLCBDE* operon encode proteins in the chorismate synthetic pathway, which is

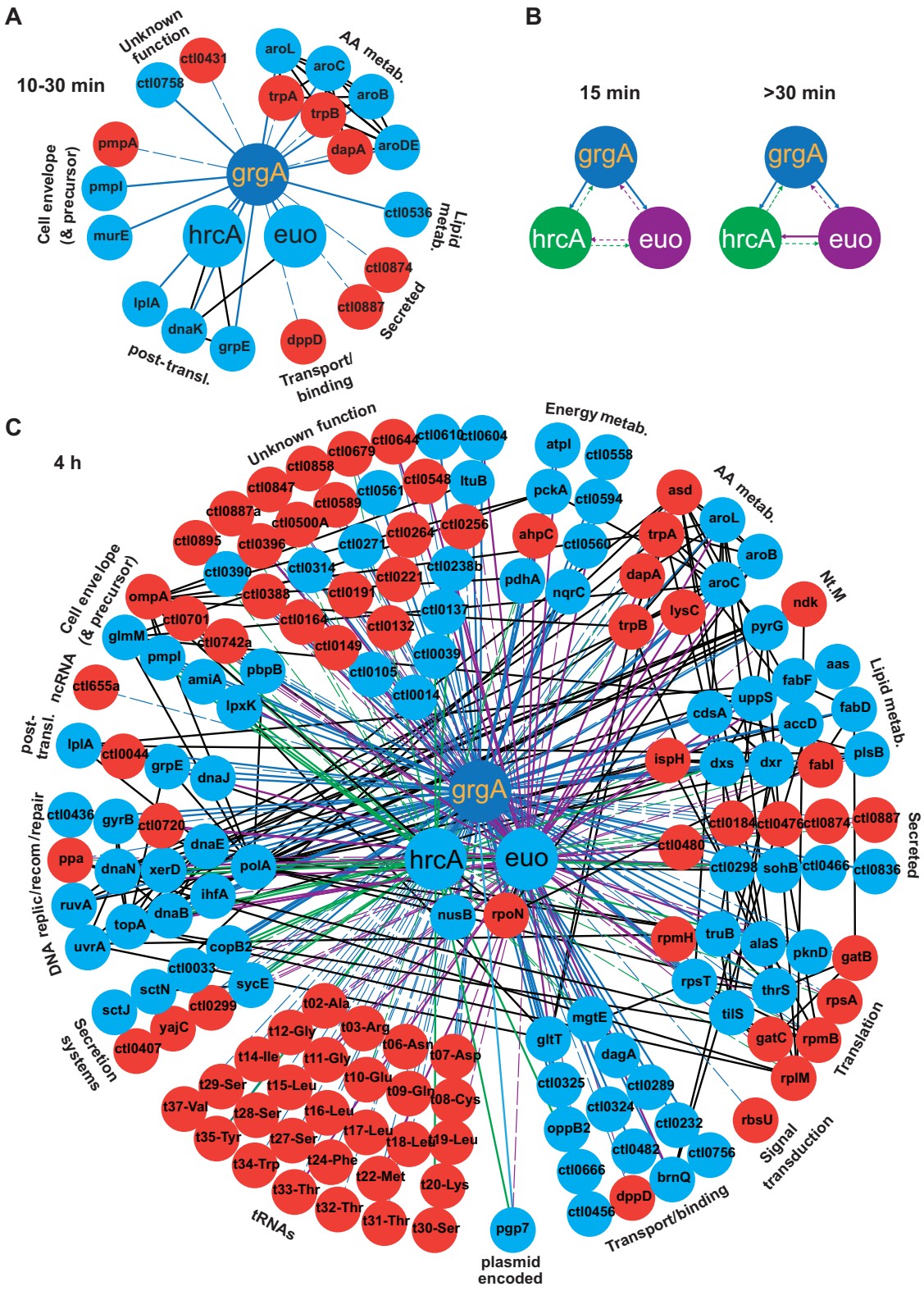

**FIG 6** GrgA-regulated TRN. (A) TRN established within 10 to 30 min of ATC-induced GrgA overexpression. Data Set S5A, which contains RNA-seq data extracted from Data Set S3 (biological triplicates), and qRT-PCR data (biological triplicates) (Fig. 3), were used to generate the TRN using STRING v11. Light blue and red nodes are genes upregulated and downregulated by GrgA, respectively. Black lines connect functional or physical associations identified by the STRING v11 database. Solid and dashed lines connect GrgA to upregulated and downregulated genes, respectively. (B) Interregulatory relationship among *grgA*, *euo*, and *hrcA* deduced following 15-min and 30-min ATC treatments of transformants. qRT-PCR data obtained from biological triplicates of CtL2/GrgA (Fig. 3B and F), CtL2/Euo (Fig. S3A), and CtL2/HrcA (Fig. S3B) were used to manually generate the TRNs. Arrows signify

upstream of the tryptophan synthetic pathway catalyzed by *trpB* and *trpA*. Paradoxically, expression of the *tribal* operon is downregulated in RNA-seq (Fig. 6A and Data Set S5C).

To determine whether Euo and/or HrcA directly regulate *grgA* expression and whether *euo* and *hrcA* directly regulate each other's expression, we performed qRT-PCR analysis for CtL2/Euo and CtL2/HrcA treated without ATC or with ATC for 15 and 30 min. In ATC-treated CtL2/Euo, *grgA* was progressively downregulated from 15 to 30 min, whereas *hrcA* was transiently downregulated at 15 min before it returned to the basal level (Fig. S3A). In ATC-treated CtL2/HrcA, *grgA* and *euo* were consistently downregulated at both 15 and 30 min (Fig. S3B). Combined with findings from previous RNA-seq (Fig. 2A and Data Set S3) and qRT-PCR analyses (Fig. 3B and F), the new qRT-PCR data in Fig. S3 revealed an interregulatory network among *grgA*, *euo*, and *hrcA* (Fig. 6B). While GrgA overexpression consistently upregulated both *euo* and *hrcA*, Euo overexpression demonstrated dynamic effects on *hrcA*. Accordingly, Euo overexpression led to an initial *hrcA* downregulation but then a complete reversal (Fig. 6B and Fig. S3A). HrcA overexpression consistently resulted in downregulation of both *grgA* and *euo* (Fig. 6B and Fig. S3B).

A GrgA TRN at 1 h of ATC treatment was developed using the RNA-seq data obtained from cultures treated with ATC for 1 h (Data Set S3). In contrast to the TRN developed at 30 min of ATC treatment, a larger number of additional genes were differentially regulated at 1 h. These genes can be classified into 17 functional groups (Fig. S4 and Data Set S5B). Further, a combined TRN in Fig. 6C was generated from differentially expressed genes detected by RNA-seq of CtL2/GrgA, CtL2/Euo, and CtL2/HrcA induced with 10 nM ATC for 4 h (Fig. 6C and Data Set S5C). The most striking takeaway from this combined TRN is the downregulation of 28 tRNAs. Of these 28 tRNAs in CtL2/GrgA treated with ATC, 7 were also downregulated in CtL2/Euo treated with ATC. However, four tRNAs downregulated by GrgA overexpression were upregulated by Euo overexpression (Fig. 6C and Data Set S5C).

In summary, network analysis suggests that the direct regulon of GrgA is comprised of 13 genes, which include *euo* and *hrcA* (Fig. 6A), and the indirect regulon of GrgA includes approximately 150 genes from a variety of functional groups (Fig. 6A and C and Fig. S4). Furthermore, our analysis indicates the existence of a dynamic interregulatory network among *grgA*, *euo*, and *hrcA* (Fig. 6B and C). Interactive versions of Fig. 6A and C and Fig. S4 can be accessed at the Figshare public data repository (https://doi.org/10.6084/m9.figshare.14195447).

**Differential developmental expression patterns for GrgA, Euo, and HrcA.** The interregulation among GrgA, Euo, and HrcA (Fig. 6) discussed earlier prompted us to determine the temporal expression pattern of each TF during the developmental cycle in wild-type CtL2 grown in L929 cells. Relative genome copy numbers were used to normalize the expression levels (Fig. 7 and Fig. S5A). qRT-PCR analysis revealed a low level of *grgA* mRNA from 0 to 5 hpi, which started to increase at 8 hpi before reaching a peak at 18 hpi (Fig. 7 and Fig. S5B). Sandwich enzyme-linked immunosorbent assay (ELISA) detected a similar temporal GrgA expression pattern (Fig. 7 and Fig. S5C). Compared to the GrgA expression pattern, a marked increase in the level of *euo* mRNA was detected at 1 hpi. The level peaked at 8 hpi, gradually decreasing through 18 hpi before reaching a nadir at 24 hpi that was maintained through the end of the culture

**FIG 6 Legend (Continued)**
direction of regulation. Solid and dotted line indicate upregulation and downregulation, respectively. (C) TRN developed by 4 h of ATC-induced GrgA overexpression. Data Set S5C, which contains a subset of RNA-seq data extracted from Data Set S2B (biological duplicates of CtL2/GrgA), Data Set S4B (biological duplicates of CtL2/Euo), and Data Set S4C (biological duplicates of CtL2/HrcA), were used to generate the TRN using STRING v11. See panel A for information for node colors and black lines. Solid and dashed lines connect TFs to upregulated and downregulated genes, respectively. Blue, purple, and green lines connect GrgA, Euo, and HrcA, respectively, to the genes that the TFs regulate. (A and C) Functional groups are labeled. Abbreviations: AA, amino acid; metab., metabolism; Nt.M, nucleotide metabolism; post-transl., posttranslational protein modification; recom, recombination; replic, replication. An interactive version of this figure is available at the Figshare public data repository (https://doi.org/10.6084/m9.figshare.14195447).

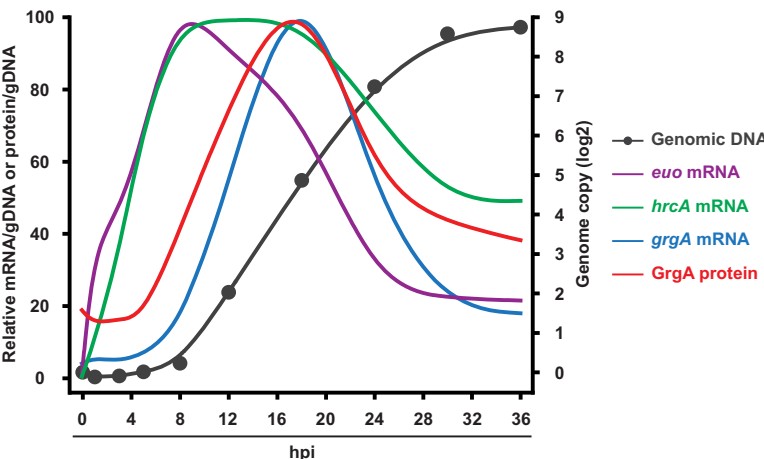

**FIG 7** Genome copy numbers and temporary expression patterns of endogenous GrgA, Euo, and HrcA during the CtL2 developmental cycle. L929 cells infected with wild-type CtL2 at the multiplicity of infection of 0.5 inclusion-forming unit per cell were harvested at 0, 1, 3, 5, 8, 12, 18, 24, 30, and 36 hpi. Genomic DNA (gDNA) was quantified using qPCR (biological triplicates). RNAs of *grgA*, *euo*, and *hrcA* were quantified using qRT-PCR (biological triplicates). GrgA protein was quantified using ELISA (biological duplicates). All expression data were normalized with the amount of gDNA. See Fig. S5 for genome copy number data and expression results of individual genes with error bars.

period (36 hpi) (Fig. 7 and Fig. S5D). A similar increase in the *hrcA* mRNA was also detected at 1 hpi, although the magnitude of change was less than that observed during *euo* mRNA detection. Similar to *euo* mRNA, levels of *hrcA* mRNA reached a peak around 8 hpi. This high level of HrcA expression persisted through 18 hpi before decreasing gradually thereafter (Fig. 7 and Fig. S5E). The temporal expression patterns of the three TFs in wild-type CtL2 supports the notion that the GrgA-Euo-HrcA TRN is crucial for chlamydial developmental and growth, as will be discussed below.

**Effects of GrgA, GrgA$\Delta\sigma^{66}$BD, GrgA$\Delta\sigma^{28}$BD, Euo, and HrcA overexpression on *C. trachomatis* growth.** The apparent roles that GrgA plays in *C. trachomatis* transcriptomic expression predict that GrgA overexpression would lead to a chlamydial growth defect. Indeed, treatment of CtL2/GrgA grown in L929 cells with 10 nM ATC between 0 and 35 hpi resulted in apparently smaller inclusions with lower intensities of a red fluorescence protein (RFP) encoded by the mKate gene of the transformed plasmid (Fig. 8A). However, identical treatment of CtL2/GrgA$\Delta\sigma^{66}$BD did not cause any observable effects on these growth markers (Fig. 8A). ATC treatment of CtL2/GrgA$\Delta\sigma^{28}$BD reduced RFP intensities, albeit the reduction was much less severe than that in CtL2/GrgA (Fig. 8A). We also demonstrated that treatment of CtL2/GrgA with 10 nM ATC starting at 12 hpi resulted in a 35% reduction in the genome copy number at 16 hpi (Fig. 8B). These findings suggest that delicate regulation of physiological GrgA concentrations is required for adequate CtL2 growth. Further, $\sigma^{66}$-dependent transcriptomic changes are absolutely required for GrgA overexpression-induced inhibition, whereas $\sigma^{28}$-dependent changes also play a significant role.

To probe the contributions of upregulated Euo and HrcA expression to GrgA overexpression-induced growth inhibition, we determined the effects of lower ATC concentrations on genome replication in CtL2/Euo and CtL2/HrcA, respectively. Treatment with subnanomolar ATC from 12 to 16 hpi increased *euo* and *hrcA* RNAs nearly 4-fold (Fig. 8C). These degrees of induction were comparable to the *euo* and *hrcA* RNA increases induced following 10 nM ATC treatment of CtL2/GrgA cultured in L929 cells (Fig. 3 and Data Set S2). In parallel, we observed a significant 40% decrease in the genome copy in the CtL2/Euo, and a statistically insignificant 10% decrease in the genome copy in the CtL2/HrcA (Fig. 8D). These data support the notion that dysregulated *euo* expression contributes to *grgA* overexpression-induced chlamydial growth inhibition, although an influence of upregulated *hrcA* expression cannot be ruled out due to the relatively short (4 h) ATC treatment duration.

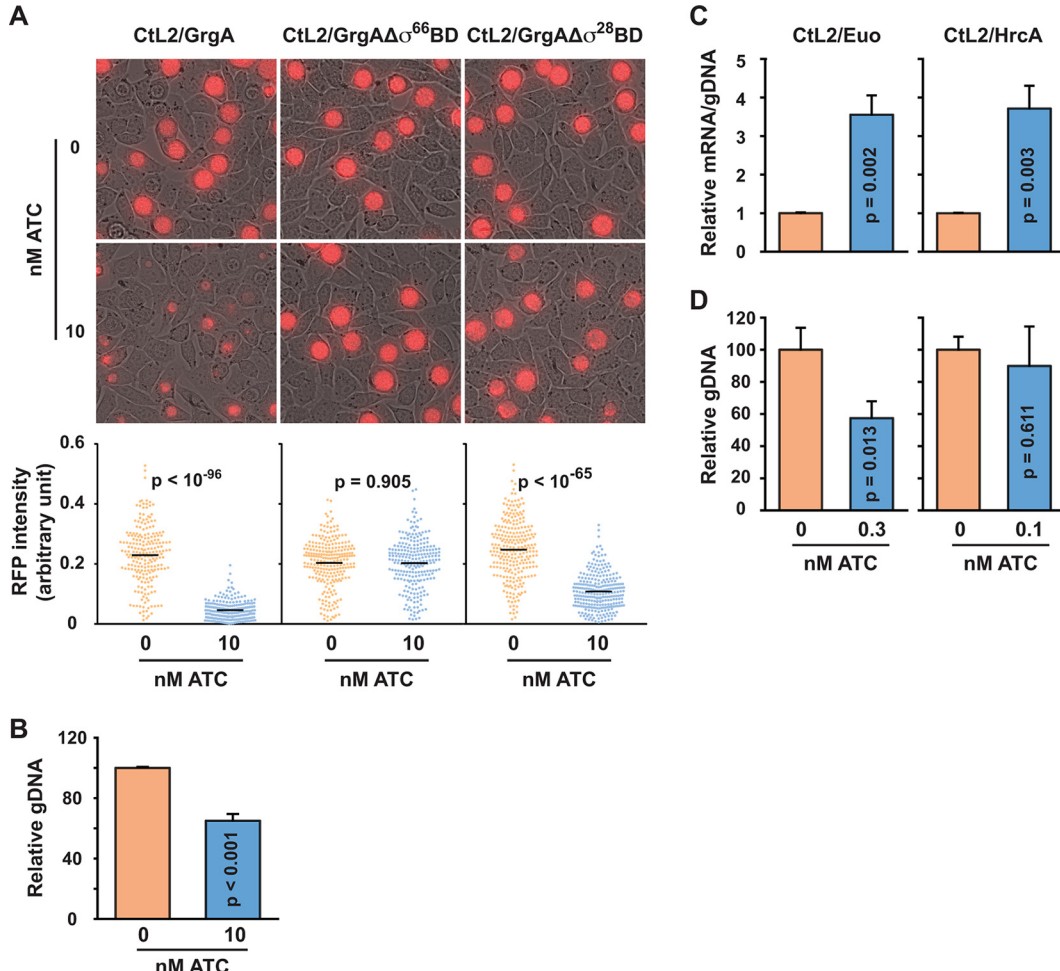

**FIG 8** Effects of overexpression of GrgA forms, Euo and HrcA on CtL2 growth in L929 cells. (A) Differential effects of 10 nM ATC treatment on inclusion morphology and RFP intensity in cultures of CtL2/GrgA, CtL2/GrgAΔσ66BD, and CtL2/GrgAΔσ28BD. ATC was added at 0 hpi. Images were acquired at 35 hpi. (B) Genome copy reduction in CtL2/GrgA after treatment with 10 nM ATC for 4 h (from 12 to 16 hpi). Genome copy was determined using qPCR. (C) Moderate increases in *euo* and *hrcA* expression in CtL2/Euo and CtL2/HrcA, respectively, after treatment with a low concentration of ATC for 4 h (from 12 to 16 hpi). RNAs of *euo* and *hrcA* were quantified using qRT-PCR. Note that mRNA expression values have been normalized with gDNA due to (possible) overexpression-induced growth inhibition (D) Genome copy reduction in CtL2/Euo but not CtL2/HrcA after treatment with a low concentration of ATC for 4 h. Genome copy was determined using qPCR. In panels B to D, all quantitative data were obtained with biological triplicates.

**GrgA overexpression-mediated CtL2 growth inhibition and upregulation of *euo* and *hrcA* in infected human cells.** All transcriptomic and growth data presented above were obtained using mouse fibroblast L929 cells, a commonly used cellular system for studying *C. trachomatis* (e.g., references 3, 9, 14, 50, and 60). We next determined the effects of GrgA overexpression on *euo* expression, *hrcA* expression, and chlamydial growth using human vaginal carcinoma HeLa cells as the host. As in L929 cells (Fig. 3B and F), we observed statistically significantly upregulated *euo* and *hrcA* transcript levels in HeLa cells following 10 nM ATC treatment for 30 min (Fig. 9A). As was observed in L929 cells (Fig. 8A), we also detected chlamydial growth inhibition in HeLa cells following 10 nM treatment for 35 h (Fig. 9B). These findings indicate that GrgA-mediated transcriptional regulation is conserved with either human or mouse cells as the CtL2 host.

## DISCUSSION

In this report, we documented the effects of full-length GrgA and domain deletion mutant overexpression on *C. trachomatis* transcriptomic expression. We identified

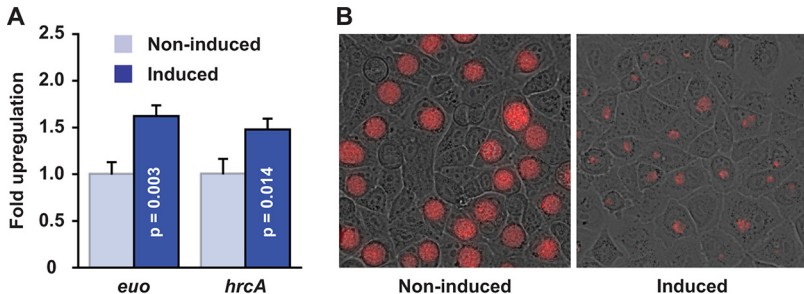

**FIG 9** Upregulated *euo* and *hrcA* expression and growth inhibition in CtL2/GrgA cultured in human HeLa cells following ATC treatment. (A) Increased *euo* and *hrcA* RNA levels following treatment with 10 nM ATC for 30 min (from 15.5 to 16 hpi). RNAs of *euo* and *hrcA* in triplicated cultures were quantified using qRT-PCR. (B) Reduced inclusion size and RFP intensity of inclusions formed by CtL2 treated with 10 nM ATC from 0 to 35 hpi.

putative direct and indirect regulons of GrgA and uncovered a TRN that encompasses GrgA, Euo, and HrcA. We also revealed temporal expression profiles for *grgA*, *euo*, and *hrcA*. Our findings have important implications for the regulation of chlamydial gene expression and progression of the chlamydial developmental cycle.

**Direct and indirect regulons of GrgA.** Our RNA-seq and qRT-PCR analyses predict putative direct and indirect regulons of GrgA by detecting time-dependent transcriptomic changes following ATC-induced GrgA overexpression. The direct regulon likely includes 13 genes that are upregulated within 10 to 30 min of ATC treatment (Fig. 3 and 6A). Chromatin immunoprecipitation followed with high-throughput DNA sequencing would be most useful for elucidating the GrgA regulon. In the absence of supporting chromatin immunoprecipitation (ChIP) data, whether or not all 13 genes are directly upregulated by GrgA requires further validation.

Evidence presented in this report reaffirms that GrgA activates both $\sigma^{66}$- and $\sigma^{28}$-dependent promoters through direct interactions with the two sigma factors, a notion drawn from our previous *in vitro* studies (39–41). Full-length GrgA overexpression caused numerous transcriptomic changes, while GrgA$\Delta\sigma^{66}$BD overexpression resulted in upregulation of only *pmpI* expression and downregulation of only four genes (Fig. 1 and Data Set S2). Additionally, among seven promoter regions upstream of the four nonoperon genes and three operons upregulated within 10 to 30 min of ATC treatment, three have both recognizable −10 and −35 elements of the $\sigma^{66}$-dependent promoter, while one has noticeable −10 and −35 elements of the $\sigma^{28}$-dependent promoter. Finally, the transcription activities of two of the $\sigma^{66}$-dependent promoters (*euo* and *hrcA*) and the $\sigma^{28}$-dependent *pmpI* promoter are stimulated by GrgA *in vitro* (Fig. 4).

Eight genes are downregulated 30 min after ATC treatment (Fig. 6A; see also Data Sets S3 and S5A in the supplemental material). These genes are likely subject to indirect rather than direct repression by GrgA. A bacterial TF can activate one of two neighboring genes and repress the other if the genes share an intergenic promoter region (61). Because none of the eight GrgA-downregulated genes is located next to a upregulated gene, the opportunity for direct repression is unlikely. Indeed, RNA-seq data from CtL2/HrcA and CtL2/Euo support possibilities that GrgA downregulates *trpB*, *trpA*, and *pmpA* expression indirectly through HrcA (Data Set S4C) and downregulates *ctl0887* and *ctl0874* expression through Euo and/or HrcA (Data Set S4B and S4C). Nonetheless, we cannot completely rule out the possibility that GrgA acts as a repressor since some bacterial TFs have dual functions (62, 63).

Excluding the 13 putative direct target genes, we consider almost all genes whose RNA levels statistically significantly increased or decreased between 1 h and 4 h as constituents of the indirect regulon of GrgA. It is important to point out that our approach of classifying direct and indirect target genes based on the induction time needed for an increase to be observed could be an overly simplistic approach because it precludes

the recognition of direct target genes whose promoters have low affinities for GrgA and thus require longer time to be activated.

Because GrgA is an activator of *euo* and *hrcA*, GrgA's indirect regulon is expected to be much larger than its direct regulon. Nearly 150 target genes were identified by 4 h when an arbitrary threshold of 1.33-fold change was applied. This approximation likely underestimates the true size of the regulon for a number of considerations. Technically, we find that RNA-seq (Data Set S3) is not as sensitive as qRT-PCR (Fig. 3) for the detection of mRNA expression changes, a phenomenon also noted recently by Soules et al. in their CtcC transcriptomic study (14). Also, as discussed above, physiological targets that show less than 1.33-fold change are excluded.

**Chlamydial growth and development controlled by the GrgA-Euo-HrcA TRN.** RNA-seq and qRT-PCR analyses performed for CtL2/GrgA, CtL2/Euo, and CtL2/HrcA suggest that the GrgA, Euo, and HrcA regulate each other's expression. Whereas GrgA serves as an activator of both *euo* and *hrcA*, Euo likely represses *grgA* and *hrcA* directly. HrcA also represses *grgA* and *euo* (Fig. 6A and B). These observations are consistent with GrgA's role as a transcription activator and Euo and HrcA's roles as transcription repressors. However, a transient *hrcA* downregulation following Euo overexpression (Fig. 6B and Fig. S3A) highlights the highly complex nature of transcriptional regulation in the RB, which likely involves other transcriptional regulators.

The rapid and substantial increases in the RNAs of *euo* and *hrcA* in chlamydiae within 1 hpi (Fig. 7 and Fig. S5D and E) suggest that Euo and HrcA as well as GrpE and DnaK (other products of the *hrcA-grpE-dnaK* operon) play critical roles in immediate early chlamydial development. Even though *grgA* is not transcribed until 8 hpi, we hypothesize that the GrgA protein prepackaged within EBs is critical for the activation of *euo* and *hrcA* immediately after EBs enter host cells, based on the interconnected relationship among *grgA*, *euo*, and *hrcA* (Fig. 6). Although our GrgA protein expression curve indicates that on average, an RB at 18 hpi (when the GrgA level peaks) contains about 5 times more GrgA molecules than an EB (Fig. 7 and Fig. S5C), the concentration of GrgA in the invading EB would be about 8 times higher than that in the RB since the RB is nearly 40 times larger in volume (3). Previously, multiple groups also detected significant amounts of GrgA protein in EBs (39, 64, 65). Therefore, the GrgA protein expression profile supports the possibility that GrgA activates *euo* and *hrcA* after DNA decondenses during the EB-to-RB transition. The inverse correlation between Euo and GrgA expression between 8 and 18 hpi likely reflects the suppression of *grgA* by Euo, which is indicated by the interregulatory network (Fig. 6B).

The large increase in GrgA expression during the midcycle (Fig. 7) suggests that GrgA plays important roles in RB replication. Given the large number of genes in its indirect regulon (Fig. 6), GrgA likely fulfills its function as a growth regulator through balanced actions of its direct and indirect target genes with roles in biosynthesis (e.g., *polA* [DNA polymerase] and *glmM* [phosphoglucosamine mutase in peptidoglycan biosynthesis]), metabolism (e.g., *atpI* [ATP synthase]), and other processes (e.g., *pknD* [serine/threonine protein kinase]). It is conceivable that GrgA downregulates tRNA genes and other genes to coordinate the transition of RBs to EBs in later developmental stages.

We identified 8 commonly upregulated and 19 commonly downregulated genes in CtL2/GrgA, CtL2/Euo, and CtL2/HrcA undergoing growth arrest due to ATC-induced overexpression of respective TFs (Fig. 5 and Data Sets S2B and 4). Their dysregulated expression may contribute to or result from chlamydial growth inhibition. Paradoxically, four of the eight commonly upregulated genes encode proteins involved in DNA replication and repair: topoisomerase I, DNA polymerase III, DNA helicase, and a site-specific tyrosine recombinase (XerD). Interestingly, all these four genes are upregulated during gamma interferon (interferon-γ)-induced chlamydial persistence when growth is also reduced to a certain degree. Equally interesting is the observation of downregulated *trpBA* expression following GrgA overexpression (Fig. 6A and C and Data Sets S2, S3, and S5) because *trpBA* is activated in response to interferon-γ that causes tryptophan

**TABLE 2** Plasmids

| Plasmid | Description or use | Reference |
|---|---|---|
| pASK-GFP-L2-mKate2 | Shuttle vector carrying a tet-inducible GFP gene and a constitutively expressed RFP gene | Wickstrum et al. (82) |
| pTRL2Δgfp | pASK-GFP-L2-mkate2 with *gfp* deleted, used as control for TF overexpression in *C. trachomatis* | Wurihan et al. (73) |
| pTRL2-NH-GrgA | For conditionally overexpressing GrgA in *C. trachomatis* | Wurihan et al. (4) |
| pTRL2-GrgAΔ$\sigma^{66}$BD | For conditionally expressing GrgAΔ$\sigma^{66}$BD (Δ1-64) in *C. trachomatis* | Wurihan et al. (4) |
| pTRL2-GrgAΔ$\sigma^{28}$BD | For conditionally expressing GrgAΔ$\sigma^{28}$BD (Δ138-165) in *C. trachomatis* | Wurihan et al. (4) |
| pTRL2-NH-Euo | For conditionally overexpressing Euo in *C. trachomatis* | This study |
| pTRL2-NH-HrcA | For conditionally overexpressing HrcA in *C. trachomatis* | This study |
| pMT1125 | Transcriptional reporter plasmid | Wilson and Tan (58) |
| pMT1125-Peuo | *C. trachomatis euo* promoter reporter plasmid | This study |
| pMT1125-PhrcA | *C. trachomatis hrcA* promoter reporter plasmid | This study |
| pMT1125-Ppmpl($\sigma^{28}$) | *C. trachomatis pmpl* $\sigma^{28}$-dependent promoter reporter plasmid | This study |

degradation in host human cells (23, 66, 67). In addition, RNAs of *ct505* (cotranscribed with GrgA [*ct504* in *C. trachomatis* D] from the same operon) and *euo* are increased in response to interferon-γ treatment (68). We postulate that GrgA could contribute to the increased *euo* expression and the growth inhibition observed during interferon-γ-induced persistence, which would require further study to confirm (68–70).

Numerous tRNAs are downregulated following GrgA midcycle overexpression (Table 1; Fig. 6C; see also Fig. S4 and Data Sets S1, S2, S3, and S5 in the supplemental material). Among the tRNAs downregulated by GrgA overexpression, several are also downregulated by Euo overexpression, but others are upregulated by Euo overexpression (Fig. 6C and Data Sets S4 and 5). While the underlying mechanism is unclear, tRNA downregulation almost certainly contributes to growth inhibition. It is probable that expression of chlamydial tRNAs is temporarily delayed in the immediate early developmental stage and also downregulated later as RBs convert to EBs.

In summary, we have identified 13 genes that are likely direct targets of GrgA, the most recently discovered TF in *Chlamydia*. By activating expression of two major transcription factors (Euo and HrcA) and by regulating expression of numerous additional genes with functions in almost all cellular processes, GrgA acts as a master transcriptional regulator that controls chlamydial growth and development.

## MATERIALS AND METHODS

**Plasmids.** Plasmids used for this study are listed in Table 2. An NEBuilder HiFi DNA assembly cloning kit was used to generate all new plasmids. Expression vectors for Euo and HrcA were constructed by replacing the green fluorescence protein (GFP) gene in pASK-GFP-L2-mKate2 with the appropriate DNA fragments. *In vitro* transcriptional reporter plasmids pMT1125-Peuo and pMT1125-PhrcA were constructed by cloning an *euo* promoter fragment (nucleotides 835937 to 835773, GenBank accession number NC_010287.1) and an *hrcA* promoter fragment (nucleotides 768239 to 768421), respectively, into pMT1125 (58). pMT1125-Ppmpl($\sigma^{28}$) was constructed in two steps. First, the putative $\sigma^{28}$-dependent (nucleotides 321618 to 321396) and fragments were cloned into the Xbal and EcoRV sites of pMT1125 using T4 DNA ligase. Second, the guanine nucleotide inside the EcoRV site was deleted using Q5 site-directed mutagenesis kit.

Plasmids constructed were Sanger sequenced at Genscript or Psomagen to confirm sequence authenticity. For chlamydial expression vectors, sequencing analysis covered the CtL2 plasmid-encoded genes, ATC-inducible promoter, and tet repressor-coding sequence, in addition to the coding sequences of TFs or their deletion mutants. Promoter fragments and the reporter cassette in pMT1125-derived vectors were sequenced.

**CtL2 strains.** Wild-type *C. trachomatis* L2 (CtL2) (strain 434/BU) was purchased from ATCC (71). The bacterium was grown using L929 cells and Dulbecco's modified Eagle medium containing 4.5 g/liter glucose and 110 mg/liter sodium pyruvate and supplemented with fetal bovine serum (final concentration, 5%), gentamicin (10 μg/ml), and cycloheximide (1 μg/ml). CtL2/GrgA, CtL2/GrgAΔ$\sigma^{66}$BD, and CtL2/GrgAΔ$\sigma^{28}$BD were reported recently (4). CtL2/Euo and CtL2/HrcA were derived in the same manner as CtL2/GrgA (4). MD-76 gradient-purified EBs of clonal transformant populations were used for transcriptomic and growth studies (72).

**Microscopic analysis of inclusions.** L929 or HeLa cells grown on six-well plates were inoculated with EBs of transformants. Following centrifugation at 900 × *g* and room temperature for 20 min, cells were switched into above-described medium without or with 10 nM ATC and incubated at 37°C. Phase contrast and RFP images were acquired at 35 hpi (73). The Java-based ImageJ software was then used to quantify RFP intensities (4).

mSystems®

**Cellular gDNA and RNA isolation.** Induction of GrgA, GrgA domain deletion mutants, Euo, and HrcA expression in *C. trachomatis* transformant-infected L929 or HeLa cells was performed with 10 nM ATC, unless indicated otherwise. Total host and chlamydial genomic DNA (gDNA) and RNA were isolated using TRI reagent (catalog no. 93289; Sigma), which separates DNA and RNA into different phases. DNA and RNA were purified in accordance with the manufacturer's instructions (74). gDNA was dissolved in a buffer containing 0.1 M HEPES and 8 mM NaOH. These samples were stored at −20°C. RNA was dissolved in diethyl pyrocarbonate (DEPC)-treated $H_2O$ and further treated with RNase-free DNase I to eliminate residual DNA contamination. Concentrations of purified gDNA and RNA samples were determined using Qubit DNA HS (high-sensitivity) (catalog no. Q32854) and RNA HS (catalog no. Q32855) assay kits, respectively (Thermo Fisher). The resultant DNA-free RNA samples were stored at −80°C.

**Quantitative PCR (qPCR) and reverse transcription-qPCR (qRT-PCR).** Thermo Fisher QS5 qPCR machine was used for qPCR and qRT-PCR analyses to quantify relative CtL2 genome copy numbers and mRNA levels, respectively. Genomic qPCR was performed using Applied Biosystems PowerUp SYBR green master mix (catalog no. A25742; Thermo Fisher Scientific) following the manufacturer's instructions. For each reaction, 5 ng of purified total host and bacterial gDNA was used as the template. The primers for genomic qPCR were qPCR-ctl0631-F (5′CGCGCACGGTTTATTGGTTT3′) and qPCR-ctl0631-R (5′AAATAGGCCCGTGTGATCCT3′). qRT-PCR was performed using Luna Universal one-step qRT-PCR kit (catalog no. E3005E; New England BioLabs [NEB]) following the manufacturer's instructions. For each reaction, 10 ng of purified total host and bacterial RNA was used as the initial template for cDNA synthesis. All genomic and qRT-PCRs were performed in technical duplicates or triplicates. qRT-PCR data of the developmental expression study were normalized with gDNA due to increases in the number of chlamydial cells from midcycle toward late cycle points. qRT-PCR data obtained from experiments with 4-h ATC treatments were also normalized with gDNA due to overexpression-induced growth inhibition in CtL2/GrgA and CtL2/Euo. No normalization was performed for qRT-PCR data obtained from short-term (10 min to 1 h) ATC treatments because qPCR demonstrated no change in the genome copy number during the treatments.

**RNA sequencing.** Total RNA integrity was determined using Fragment Analyzer (Agilent) prior to RNA-seq library preparation. Illumina MRZE706 Ribo-Zero Gold Epidemiology rRNA removal kit was used to remove mouse and chlamydial rRNAs. Oligo(dT) beads were used to remove mouse mRNA. RNA-seq libraries were prepared using Illumina TruSeq stranded mRNA-seq sample preparation protocol, subjected to the quantification process, pooled for cBot amplification, and sequenced with the Illumina HiSeq 3000 platform with 50-bp single-read sequencing module. On average, 20 to 25 million reads were obtained for each RNA-seq sample. Short read sequences were first aligned to the CtL2 chromosome (accession number NC_010287.1) and the transformed plasmids using TopHat2 aligner, quantified for gene expression by HTSeq to obtain raw read counts per gene, and then converted to FPKM (<u>f</u>ragment <u>p</u>er <u>k</u>ilobase of gene length per <u>m</u>illion reads of the library) (75–77). DESeq, a statistical method appropriate for analyzing a small number of RNA-seq replicates (76), was used to normalize data and find group-pairwise differential gene expression based on three criteria: $P < 0.05$, average FPKM > 1, and fold change ≥ 1. Genes were clustered into groups based on temporal patterns of transcriptomics using Gaussian mixture models (48).

**TRN development.** TRNs were developed for significantly differentially regulated genes (i.e., genes with a ≥1.33-fold change, adjusted $P < 0.05$) using program STRING v11 (55). Edges for GrgA-, Euo-, and/or HrcA-regulated genes encoding hypothetical proteins were manually developed (55). Previously identified networks were automatically integrated into the GrgA TRN by STRING v11. Genes were placed into functional groups as previously described (78) with modifications (79, 80).

***In vitro* transcription assay.** Chlamydial RNA polymerase holoenzyme was partially purified from RBs of pTRL2Δgfp-transformed CtL2 using heparin agarose (Sigma) as previously described (39). *In vitro* transcription assays for $\sigma^{66}$-dependent promoters and $\sigma^{28}$-dependent promoters were performed as previously described (39, 40).

**Quantification of the GrgA protein.** Expression of the GrgA protein during the *C. trachomatis* developmental cycle was quantified using ELISA. The plates (24-well plates) were seeded with L929 cells. On each plate, four wells were inoculated with wild-type CtL2 434/BU at a multiplicity of infection (MOI) of one inclusion-forming unit (IFU) per two cells. The plates were centrifuged for 20 min at 900 × g and washed five times thereafter with 100 μg/ml heparin in HBSS to remove free and cell surface-bound EBs. They were cultured at 37°C with medium containing 1 μg/ml cycloheximide. At the intended time (0, 1, 3, 5, 8, 12, 18, 24, 30, or 36 hpi), a plate was removed from the incubator. Cells from two infected wells and two mock-infected wells were harvested in 1% sodium dodecyl sulfate (SDS) (200 μl/well) and immediately heated for 10 min at 100°C. After centrifugation (20,000 × g, 10 min), supernatants were collected and stored in aliquots at −20°C. gDNA was extracted from the remaining two infected wells and further purified as described above.

The capture antibody for the GrgA ELISAs was an antigen-affinity purified rabbit anti-GrgA antibody (81). It was diluted with 0.1 M $NaHCO_3$ to a final concentration of 2 μg/ml and added (50 μl/well) to ELISA plates (catalog no. 505021; Nest Scientific). After an overnight incubation at 4°C, the plates were washed four times with phosphate-buffered saline (PBS) to remove residual capture antibodies and blocked thereafter with 10% fetal bovine serum (FBS) at room temperature for 1 h. Samples of stored cell extracts were thawed and brought to homogeneity again by gentle pipetting. Cell extracts harvested between 0 to 8 hpi were diluted with equal volume of $H_2O$ to yield 0.5% SDS after dilution. Those harvested at 12 hpi were diluted fivefold using 0.3% SDS. Those harvested at 18, 24, 30, and 36 hpi were diluted 150-, 500-, 1,500- and 1,500-fold, respectively, with 0.5% SDS. Diluted cell extracts in triplicate were added to ELISA plates (100 μl/well) after removal of FBS and two washes. The plates were incubated on a shaker for 1 h at room temperature (RmT), washed five times with PBS, and sequentially

reacted with 1:1,000 diluted polyclonal mouse anti-GrgA (39), and 1:1,000 diluted peroxidase-conjugated goat anti-moue IgG (catalog no. A4416; Sigma) (1 h per reaction with five washes after each). Developing solution was prepared fresh by adding a 0.1 M TMB (3,3′,5,5′-tetramethylbenzidine) solution (catalog no. 806336; Sigma) and 3% $H_2O_2$ with a phosphate citrate buffer (0.2 M $Na_2HPO_4$, 0.1 M citric acid [pH 5.0]) to yield final concentrations at 2.5 mM and 0.02% $H_2O_2$, respectively. The developing solution was added onto the ELISA plates (100 $\mu$l/well). After 30-min incubation at RmT, 100 $\mu$l of 2 N $H_2SO_4$/well was added to stop the reaction. Absorbance at 450 nm was obtained on a Spectramax iD5 plate reader (Molecular Devices). The concentrations of GrgA in the infected-cell extracts were calculated using standard curves obtained with 0 to 1,000 pg/ml purified recombinant GrgA (39, 40) after removing signals generated with mock-infected cell extracts. Genome copy numbers obtained by qPCR for corresponding time points were used to normalize the GrgA expression level.

**Statistical analysis.** qRT-PCR, *in vitro* transcription data, and RFP intensity were analyzed using *t* tests in Excel of Microsoft Office. When applicable, *P* values were adjusted for multiple comparisons by Benjamini-Hochberg procedure to control the false discovery rate.

**Data availability.** RNA-seq data have been deposited into the NCBI Gene Expression Omnibus under accession number GSE153747.

## SUPPLEMENTAL MATERIAL

Supplemental material is available online only.
**DATA SET S1,** XLSX file, 0.2 MB.
**DATA SET S2,** XLSX file, 0.5 MB.
**DATA SET S3,** XLSX file, 0.6 MB.
**DATA SET S4,** XLSX file, 0.4 MB.
**DATA SET S5,** XLSX file, 0.1 MB.
**FIG S1**, PDF file, 1.4 MB.
**FIG S2**, PDF file, 0.4 MB.
**FIG S3**, PDF file, 0.5 MB.
**FIG S4**, PDF file, 0.6 MB.
**FIG S5**, PDF file, 0.3 MB.

## ACKNOWLEDGMENTS

We thank our colleagues Joseph Fondell and Wei Vivian Li for discussions and P. Scott Hefty (University of Kansas) for the supply of pASK-GFP/mKate2-L2.

This work was supported by grants from the National Institutes of Health (grants AI122034, AI140167, and AI154305 to H.F.) and New Jersey Health Foundation (grant PC98-20 to H.F.). The Genome Sequencing Facility at the University of Texas Health San Antonio (UTHSA) is supported by NIH-NCI P30 CA054174 (Cancer Center at UT Health San Antonio), NIH Shared Instrument grant OD021805 (S10 grant), and CPRIT Core Facility Award (RP160732). Y.H. was supported by a scholarship from the China Scholarship Council (CSC) from September 2018 to September 2020.

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
