## [Reviewer comments · mSystems]

Identification of A GrgA-Euo-HrcA transcriptional regulatory network in Chlamydia

Wurihan Wurihan, Yi Zou, Alec Weber, Korri Weldon, Yehong Huang, Xiaofeng Bao, Chengsheng Zhu, xiang wu, Yaqun Wang, Zhao Lai, and Huizhou Fan

Corresponding Author(s): Huizhou Fan, Rutgers University Robert Wood Johnson Medical School

Review Timeline:

Submission Date:

June 23, 2021

Accepted:

July 6, 2021

Editor: Ryan McClure

Reviewer(s): The reviewers have opted to remain anonymous.

Transaction Report:

DOI: <https://doi.org/10.1128/mSystems.00738-21>

July 6, 2021

Prof. Huizhou Fan
Rutgers University Robert Wood Johnson Medical School
Department of Pharmacology
683 Hoes Lane West
Piscataway, NJ 08854

Re: mSystems00738-21 (Identification of A GrgA-Euo-HrcA transcriptional regulatory network in Chlamydia)

Dear Prof. Huizhou Fan:

Your manuscript has been accepted, and I am forwarding it to the ASM Journals Department for publication. For your reference, ASM Journals' address is given below. Before it can be scheduled for publication, your manuscript will be checked by the mSystems senior production editor, Ellie Ghatineh, to make sure that all elements meet the technical requirements for publication. She will contact you if anything needs to be revised before copyediting and production can begin. Otherwise, you will be notified when your proofs are ready to be viewed.

As an open-access publication, mSystems receives no financial support from paid subscriptions and depends on authors' prompt payment of publication fees as soon as their articles are accepted. =

Publication Fees:

We recognize that the video files can become quite large, and so to avoid quality loss ASM suggests sending the video file via <https://www.wetransfer.com/>. When you have a final version of the video and the still ready to share, please send it to Ellie Ghatineh at eghatineh@asmusa.org.

Sincerely,

Ryan McClure
Editor, mSystems

Journals Department
sDataset 3: Accept
sFig. 5: Accept
sFig. 3: Accept
sFig. 2: Accept
sDataset 2: Accept
sDataset 1: Accept
sFig. 4: Accept
sDataset 4: Accept
sFig. 1: Accept
sDataset 5: Accept